**Data Availability Statement:** All data underlying our analysis are publicly available at the project repository at https://github.com/lhncbc/CDE/tree/master/hiv/datadictionary.

**Funding:** This research was supported by the NIH Office of AIDS Research and Intramural Research

# Analysis of data dictionary formats of HIV clinical trials

**Craig S. Mayer**⬤*◉, **Nick Williams**◉, **Vojtech Huser**◉

Lister Hill National Center for Biomedical Communication, National Library of Medicine, NIH, Bethesda, MD, United States of America

◉ These authors contributed equally to this work.
\* craig.mayer2@nih.gov

## Abstract

### Background

Efforts to define research Common Data Elements try to harmonize data collection across clinical studies.

### Objective

Our goal was to analyze the quality and usability of data dictionaries of HIV studies.

### Methods

For the clinical domain of HIV, we searched data sharing platforms and acquired a set of 18 HIV related studies from which we analyzed 26 328 data elements. We identified existing standards for creating a data dictionary and reviewed their use. To facilitate aggregation across studies, we defined three types of data dictionary (data element, forms, and permissible values) and created a simple information model for each type.

### Results

An average study had 427 data elements (ranging from 46 elements to 9 945 elements). In terms of data type, 48.6% of data elements were string, 47.8% were numeric, 3.0% were date and 0.6% were date-time. No study in our sample explicitly declared a data element as a categorical variable and rather considered them either strings or numeric. Only for 61% of studies were we able to obtain permissible values. The majority of studies used CSV files to share a data dictionary while 22% of the studies used a non-computable, PDF format. All studies grouped their data elements. The average number of groups or forms per study was 24 (ranging between 2 and 124 groups/forms). An accurate and well formatted data dictionary facilitates error-free secondary analysis and can help with data de-identification.

### Conclusion

We saw features of data dictionaries that made them difficult to use and understand. This included multiple data dictionary files or non-machine-readable documents, data elements included in data but not in the dictionary or missing data types or descriptions. Building on

Program of the National Institutes of Health (NIH)/ National Library of Medicine (NLM)/Lister Hill National Center for Biomedical Communications (LHNCBC) Funder: NIH Office of AIDS Research (https://www.oar.nih.gov/) The funders had no role in study design, data collection and analysis, decision to publish, or preparation of the manuscript.

**Competing interests:** NO authors have competing interests

experience with aggregating data elements across a large set of studies, we created a set of recommendations (called CONSIDER statement) that can guide optimal data sharing of future studies.

# 1 Introduction

In recent years, efforts to define research Common Data Elements (CDEs) attempt to harmonize data collection across clinical studies [1]. Sheehan at al. defined a CDE as 'a combination of a precisely defined question paired with a specified set of responses to the question that is common to multiple datasets or used across different studies' [2]. CDEs have been defined on both a general level applicable to a broad range of diseases and studies, as well as on a disease specific level that focuses on data elements applicable to a narrow context. An example of general data elements are those defined by PhenX initiative, such as employment status, education attainment, or health insurance coverage (all with appropriate permissible values for such elements). An example of disease specific CDEs are those defined by the Therapeutic Area User Guide for HIV [3] published by the Clinical Data Interchange Standard Consortium (CDISC). CDEs are expected to deliver the following benefits: faster and cheaper study start-up, improved comparability and aggregation of data across studies, improved study data collection and study data quality, and improved data organization for re-use.

Another phenomenon that highlights the importance of CDEs is the requirement to share de-identified individual participant data (IPD) of a completed observational study or interventional trial [4]. When data is shared, a data dictionary is typically provided to describe individual data elements used in a study. In recent decades, tens of new data sharing platforms with the aim of facilitating secondary research have emerged [5].

Identification and standardization of CDEs is, however, an ongoing challenge in the field of clinical research informatics [6]. Some leaders have argued strongly for adoption of policies that require much stronger sponsor-enforced standardization [2, 7], while others point to possible restrictions and additional CDE burden on principal investigators of future studies [8, 9]. Increased pressure for CDE adoption can be seen in research efforts triggered by the COVID-19 pandemic [10, 11]. Many CDE initiatives use expert consensus to achieve standardization. We refer to this method as a top down approach. If the number of data elements to standardize is large, expert consensus can be very time consuming. With increased availability of de-identified IPD data from completed studies, it is possible to arrive at CDEs using a data-driven approach. Most common data elements will simply appear in a high number of shared study datasets if a simple usage frequency approach is used. We refer to this method as a bottom up approach. A data-driven approach can also handle a large volume of common data elements. However, this data-driven approach depends on studies sharing their data elements in an analyzable format.

These two general clinical research trends (effort to arrive at standardized CDEs and increased sharing of completed studies) have also impacted the clinical domain of infectious diseases and HIV. For example, at ClinicalTrials.gov, the number of studies that provide a link to de-identified IPD increased from 4.4% in 2014 (3 HIV studies have link to IPD out of 68 total HIV studies that completed in 2014) to 12.6% in 2019 (8 HIV studies have link to IPD out of 63 total HIV studies that completed in 2019). Such numbers demonstrate a gradual shift towards increased data sharing.

Despite improvements, there are still opportunities for further enhancements in terms of format and the extent of data sharing in the HIV domain as well in clinical research in

general). We present a study to analyze the quality and usability of data dictionaries of HIV studies. We focused on data dictionaries because they are an important common metadata artifact for data sharing. We thus search for HIV/AIDS interventional trials or observational studies using several approaches. In our article, we use the term study to refer to both interventional trials and observational studies. Our goal is to find HIV studies that provide a data dictionary with a list of data elements used in a study and relevant additional metadata. We analyze the format and content of those data dictionaries. The study contribution lies in the generating of recommendations that improve data sharing of future HIV studies. Our study is the first informatics study of data sharing format that analyzes a large body of HIV studies shared to date via various data sharing mechanisms. The presented study is limited to data dictionary analysis, although the motivation is to later analyze a large body of past HIV data elements to inform data-driven consensus on CDEs. This study is part of a larger research project titled 'Identification of Research Common Data Elements in HIV/AIDS using data science methods' [12].

## 2 Materials and methods

In this section, we outline how we identified a set of analyzed HIV studies and define important terms for the analysis of data dictionaries. We also describe relevant standards for study data dictionaries.

### 2.1 Trials acquisition

We searched clinical study data sharing platforms [5] for HIV studies that shared IPD. We defined HIV studies as any study relating to HIV, which includes any study with HIV positive participants, or any study relating to HIV infection (i.e. HIV prevention or vaccine trials). A data sharing platform is a web-based repository of completed clinical studies with study IPD and other study artifacts (such as protocol, study publications or case report forms) for each included study. The platforms we searched included National Institute of Drug Abuse (NIDA) Data Share [13], the National Heart, Lung and Blood Institute (NHLBI) BioLINCC, the National Institute of Child Health and Human Development Data and Specimen Hub (NICHD DASH), Vivli, clinicalstudydatarequest.com, Project Data Sphere, the National Institute of Mental Health Data Archive (NDA) and the Yale Open Data Access Project. We also searched for HIV clinical trial networks using a web search engine and requested IPD from completed studies conducted by those networks. We excluded studies for which a data dictionary could not be obtained or inferred from IPD or studies that were not registered at Clinical-Trials.gov. In order to aggregate this data across many HIV studies, we obtained Institutional Review Board approval at our local institution (or exemption if the data were de-identified). We filed appropriate data requests at the platforms where we found relevant HIV studies.

### 2.2 Data dictionaries

For each HIV study included in our analysis, we organized the available data artifacts into the following categories: IPD data files, data dictionary files, study protocol documents and study de-identification notes. Within the data dictionary category, we observed several different formats. Some studies used single or multiple Comma Separated Value files (CSV), a widely accessible and machine-readable file format. Some studies used Portable Document Format (PDF) or a proprietary SAS data catalog format (*.sas7bcat) [14]. Because we used computerized methods for aggregation, we did not include studies whose data dictionaries were not in a machine-readable format or could not be readily converted into such a format. This includes PDFs with scanned content.

In our analysis of data dictionaries and data elements, we adopt the definitions developed by NIH CDE Task Force of the NIH Data Council [15]. They clearly defined terms such as data element, common data element, form and permissible value [16]. In addition to using terminology defined by the NIH CDE task force, we have introduced three types of data dictionaries.

We use the term Element Data Dictionary (or element dictionary in shorter form) to describe a spreadsheet that enumerates individual data elements used in a study with fields such as data element name and description. We combined all data element data dictionaries to create an aggregate data elements file that contains data elements from a set of studies. Due to the large number of data elements and to limit the project scope, we did not perform any semantic mapping of identical DEs. This aggregated file targeted the following attributes about each data element (referred to as target DE model): (1) data element description, (2) data element ID (sometimes called DE name, DE short label, variable name, or variable ID), (3) data type (such as character, numeric, date, enumerated, or boolean), (4) length (provides information about the length of the character string or maximum value or range for a numeric data element), and (5) group ID (sometimes called form name or form ID; provides information on which form the DE is being collected). To demonstrate some examples of study data elements, Table 1 shows the element data dictionary from study NCT00099359: 'Trial of Three Neonatal Antiretroviral Regimens for Prevention of Intrapartum HIV Transmission.' For brevity, the table shows only 10 selected data elements out of all 577 data elements used in that study.

For each element data dictionary, we analyzed the accuracy and completeness of the dictionary. This included an analysis of the previously specified features, such as the presence and clarity of the data element descriptions and the prevalence, common usage, and accurate representation of the data type for each listed data element.

We use the term Forms Data Dictionary (or forms dictionary in shorter form) to refer to a data dictionary that provides a full list of titles and descriptions of all Case Report Forms (CRFs) used in the study (or other relevant metadata for data element grouping). Table 2 shows an example of a forms dictionary from study NCT01751646: 'Vitamin D Absorption in HIV Infected Young Adults Being Treated With Tenofovir Containing cART.' We followed the same approach as with the element data dictionary and combined all form dictionaries across all analyzed studies. The intention was to see whether some CRF titles appear more frequently. To eliminate similarly named forms, we manually mapped synonymous form titles to their preferred title and identified common CRF names that appeared in at least two studies. For example, forms by the names 'F89' for NCT01751646 and 'hxw0100' for NCT01418014 both map to a common form name of 'family history'. The semantic mapping was much more

**Table 1. Example data elements (for trial NCT00099359: 'Trial of Three Neonatal Antiretroviral Regimens for Prevention of Intrapartum HIV Transmission').**

| Data Element Description | Data Element ID | Data Type | Length | Group ID (= Form Name) |
|---|---|---|---|---|
| Antiretroviral therapy during L & D? | ARTDLD | CHAR | 3 | Eligibility Questionnaire |
| Birth Weight (grams) | BRTWT | NUMBER | 4 | Eligibility Questionnaire |
| Derived Age | DAGE | NUMBER | 5 | Eligibility Questionnaire |
| HIV Method | HIVMTD | CHAR | 66 | Eligibility Questionnaire |
| Is the subject HIV infected? | HIVINF | CHAR | 3 | Eligibility Questionnaire |
| Any Immunizations? | IMMNYN | CHAR | 7 | Infant Non-Study Medications |
| Test Value | TSTVLU | NUMBER | 12 | Laboratory Results |
| Date started | ONSETD | DATE | 8 | Medical Events |
| Specimen obtained for confirmatory test? | CNFTYN | CHAR | 3 | Syphilis Test Result |
| Years of formal education | YRSEDC | NUMBER | 2 | Maternal Demographics |

**Table 2. List of 6 selected forms (out of 40 total) present in the forms dictionary for NCT01751646: 'Vitamin D Absorption in HIV Infected Young Adults Being Treated With Tenofovir Containing cART'.**

| Group ID | Description |
| --- | --- |
| B100 | Specimen Tracking Form |
| BSCR | Subject Screening Log |
| C100 | Specimen Tracking Form |
| F1 | Eligibility and Enrollment Form |
| F101 | Dual Energy X-Ray Absorptiometry Form |
| F15 | Pregnancy Result Form |

feasible for forms (compared to data elements, where it was out of project scope) due to the relatively small total number of forms across all studies. For harmonizing data collection across studies, the issue of copyright protection on case report forms must be considered [17, 18]. To measure the impact of copyright, we analyzed whether any of the form (across all studies in our sample) were marked as copyrighted.

Finally, we use the term Permissible Values Data Dictionary (or Permissible Values Dictionary in shorter form) to list permissible values that are possible for categorical data elements. Each permissible value (on separate rows) is linked to the data element it provides the value for (the link is via data element ID). For example, in study NCT01772823 'An Open Label Demonstration Project and Phase II Safety Study of Pre-Exposure Prophylaxis' for data element OFFTXR, which describes the reason for discontinuation, its permissible value dictionary has 6 permissible values defined: 1 (Viral Breakthrough), 2 (Adverse Reaction), 3 (Subject's Decision), 4 (Clinician's Decision), 5 (Course Completed), and 99 (Other). Each row represents a single possible value (organized under a respective data element). The permissible value information model has the following columns defined: (1) permissible value ID (or permissible value short label; this column is optional and can be missing), (2) permissible value, (3) permissible value description (if the previous field does not sufficiently define the permissible value), and (4) data element ID (for proper linking of this permissible value dictionary to the element dictionary). For the permissible value dictionaries, we assessed commonly used formats for permissible values, as well as the most commonly used values.

For all analyzed studies, we also looked at primary and secondary outcomes [1] as defined for each study at ClinicalTrials.gov. The assumption was that every outcome on ClinicalTrials.gov would be linked to at least one study data element. During our review of the study's ClinicalTrials.gov record we also determined whether the ClinicalTrials.gov study record reflects the availability of the study results data, IPD data [19] or data dictionary.

Finally, during aggregation of data dictionaries across studies, we recorded positive features of dictionaries as well as challenges of dictionary formats that complicated the analysis. Our goal was to create a set of recommendations for optimal data sharing for future HIV studies (presented in the discussion section).

## 2.3 Relevant standards

In addition to analyzing a set of real studies, we also investigated relevant standards for clinical study data dictionaries. Our reason why we looked into such standards was to inform our efforts to combine multiple data dictionaries into a single data structure. There are two relevant data dictionary standards: CDISC Define-XML and REDCap.

CDISC Define-XML standard is used by the Food and Drug Administration in the US [20] and Pharmaceuticals and Medical Devices Agency in Japan. It was first released in 2005 and it uses eXtensible Markup Language (XML) to describe a data dictionary of a clinical study.

REDCap data dictionary format is another standard defined by the REDCap electronic data capture system used by more than 3200 institutions world-wide [21]. REDCap stands for REsearch Data Capture. The REDCap software was first released in 2004 and it uses a ZIP compressed spreadsheet file to represent a data dictionary. While it is not widely used by individual studies, it is of note that, for example, the PhenX initiative provides their CDEs in the REDCap data dictionary format [22]. We also acknowledge that ISO 11179 specification aims at describing data element metadata, however it does not clearly define an exchange format [23].

## 3 Results

### 3.1 Analyzed set of studies

Our platform search identified relevant HIV studies on three data sharing platforms: NHLBI BioLINCC (3 studies), NICHD Data and Specimen Hub (DASH) (11 studies) and NIDA Data Archive (5 studies). We searched 5 additional data sharing platforms which did not yield any more results due to several reasons, including: (1) we did not find any HIV studies on those platforms, (2) the study data request process was still pending at the start of our dictionary analysis, or (3) we could not obtain the data.

Our web search identified 5 HIV/AIDS clinical trial networks. They were the HIV Vaccine Trials Network (HVTN), the HIV Prevention Trials Network (HPTN), the AIDS Clinical Trial Group (ACTG), the International Maternal, Pediatrics Adolescent AIDS and the Microbicide Trials Network (MTN). After filing official or email requests, we obtained studies from two networks: (1) HPTN (9 studies were obtained), and (2) HVTN (2 studies were obtained). One trial network, MTN, was inactive and for the two remaining networks, ACTG and IMPAACT, our data request was pending at the time of our data dictionary analysis.

Our web search for individual HIV studies (outside any network) identified one additional HIV study (Multicenter AIDS Cohort Study; MACS). Our searches were conducted between October 1, 2018 and March 31, 2019.

After pooling all possible study acquisition channels, we acquired a total of 31 HIV studies from 5 distinct sources. NIDA Data Share was the only source with a widely used standard as it used CDISC. Both HPTN and HVTN stated that newer trials in their networks would be standardizing to CDISC as well. All other data sharing platforms and studies allowed for custom formats to be used by the studies present on their platform. Of our set of studies, one from HPTN and the 5 acquired from the NIDA Data Archive used a CDISC format. Studies that used CDISC format were excluded from the main analysis presented here (we have a separate research project that focuses solely on CDISC-formatted studies). For another 7 studies in our input set, the data dictionary was either (1) not convertible into a machine-readable format, (2) the dictionary was not included in the shared data package, or (3) the dictionary could not be readily inferred from the provided IPD data. This resulted in a total of 18 studies in our final sample of studies that we analyzed. Table 3 provides a ClinicalTrials.gov study identifier and the study titles for this final set. Seven studies (38.9%) in our sample were observational studies while the remaining 11 (61.1%) were interventional trials.

### 3.2 Data element dictionaries

We identified a total of 26 328 data elements across the analyzed 18 studies. See Supplemental file S1 at the project repository, https://github.com/lhncbc/CDE/tree/master/hiv/datadictionary, to see the aggregated data element file (complete list of data elements). The median number of data elements for the studies in our set was 427 elements. The number of

**Table 3. List of 18 HIV trials in the final set analyzed for data elements.**

| NCT_ID | Title |
|---|---|
| NCT02404311* | A Safety and Immune Response Study of 2 Experimental HIV Vaccines (HVTN100) |
| NCT01772823 | An Open Label Demonstration Project and Phase II Safety Study of Pre-Exposure Prophylaxis (ATN110) |
| NCT01769456 | An Open Label Demonstration Project and Phase II Safety Study of Pre-Exposure Prophylaxis Use Among 15 to 17 Year Old Young Men Who Have Sex With Men (YMSM) (ATN113) |
| NCT01751646* | Vitamin D Absorption in HIV Infected Young Adults Being Treated With Tenofovir Containing cART (ATN109) |
| NCT01751594 | Testing a Secondary Prevention Intervention for HIV-Positive Black Young Men Who Have Sex With Men (ATN104) |
| NCT01492842 | Correlates of Oral Human Papillomavirus Infection in Adolescents and Young Adults With Behaviorally Acquired HIV (ATN114) |
| NCT01418014* | Adolescent Master Protocol (PHACS) |
| NCT01233531* | Effects of Cash Transfer for the Prevention of HIV in Young South African Women (HPTN068) |
| NCT01203332 | Identifying Undiagnosed Asymptomatic HIV Infection in Hispanic/Latino Adolescents and Young Adults (ATN096) |
| NCT00865566* | Safety and Effectiveness of HIV-1 DNA Plasmid Vaccine and HIV-1 Recombinant Adenoviral Vector Vaccine in HIV-Uninfected, Circumcised Men and Male-to-Female (MTF) Transgender Persons Who Have Sex With Men (HVTN505) |
| NCT00710593 | Impact of a Human Papilloma Virus (HPV) Vaccine in HIV-Infected Young Women (ATN064) |
| NCT00683579 | Neurocognitive Assessment in Youth Initiating HAART (ATN071) |
| NCT00491556 | Preservation and Expansion of T-cell Subsets Following HAART De-intensification to Atazanavir/Ritonavir (ATV/r) (ATN061) |
| NCT00099359* | Trial of Three Neonatal Antiretroviral Regimens for Prevention of Intrapartum HIV Transmission (HPTN040) |
| NCT00046280* | Multicenter AIDS Cohort Study (MACS) |
| NCT00005274* | Pediatric Pulmonary and Cardiovascular Complications of Vertically Transmitted HIV Infection (P2C2) |
| NCT00005273* | Pulmonary Complications of HIV Infection Study (PACS) |
| NCT00000590* | Anti-HIV Immunoglobulin (HIVIG) in Prevention of Maternal-Fetal HIV Transmission (Pediatric ACTG Protocol 185) (PACTG) |

(* indicates study for which we also obtained IPD data).

elements ranged between a minimum of 46 elements (study NCT02404311) and a maximum of 9 945 elements (study NCT00046280).

**3.2.1 Data dictionary format.** A total of 14 studies used only CSV files to provide their data dictionaries. In most cases, the studies provided a single CSV dictionary file for the entire study. A single dictionary file is the most user-friendly format for data re-using researchers. A minority of studies split the dictionary into multiple files. The most extreme case of a split dictionary was a study that provided 55 dictionary files (one for each of the 55 study data files generated).

Four studies (NCT00000590, NCT00005273, NCT00005274, and NCT01418014) used PDF files to share the element dictionary. This PDF format required a manual conversion into a CSV machine readable format. One study (NCT01233531) had a mixture of formats with some data elements provided in a PDF format and some in CSV format (spread across 8 files). For some of the studies where we also obtained IPD data (10 studies), we saw data elements present in data, but missing and not defined in the data dictionary, making the data dictionary incomplete. To quantify this level of completeness, we generated data element dictionaries for half of the trials we had IPD for and calculated the percentage of data elements included in the data dictionary. Table 4 presents the results of this dictionary completeness analysis and shows that completeness ranges from 45.4% to 100%.

**Table 4. Proportion of data elements found in IPD data that are also present in the data dictionary.**

| NCT_ID | Data Elements in Data | Included Elements in Dictionary | % of Elements in Dictionary |
|---|---|---|---|
| NCT00000590 | 179 | 179 | 100.0% |
| NCT00005273 | 1681 | 1675 | 99.6% |
| NCT01418014 | 3001 | 2979 | 99.3% |
| NCT00099359 | 389 | 337 | 86.6% |
| NCT01751646 | 945 | 429 | 45.4% |

**3.2.2 Data type.** We observed missing or incorrect information about data types within our sample of studies. Explicit declaration of data type for each data element is important for proper data interpretation and correct data analysis. In one psychology study, incorrect results were published when a categorical variable (code for country of birth) was incorrectly used as a numerical variable in the model [24]. Thanks to data sharing and secondary analysis by a researcher (external to the original study team) the error was discovered and revised study results were generated.

*3.2.2.1 Missing data type.* In terms of missing data type, 12 studies provided data type for all DEs. One study (NCT01492842) had missing data type for 49% of its data elements. Data type was completely missing in four studies (22% out of 18 studies). On an aggregate level, across all studies, a total of 10 755 DEs had missing data type (40.9% out of all 26 328 DEs). However, this was mainly due to a single study (The Multicenter AIDS Cohort Study [MACS]; NCT00046280) with 9 945 DEs without data type which accounted for 92.5% of all DEs with missing data type. The breakdown of the number of data elements for a given data type can be found in Table 5. For the 15 573 data elements where data type was declared the most common data type was string with 48.6%, followed by numeric with 47.8%.

*3.2.2.2 Incorrect data type.* In terms of incorrect data type, we observed that no studies used a categorical data type. Use of categorical variables is common in research. For example, all 13 studies in the National Sleep Research Resource [25], which is a sharing platform that we consider exemplary in terms of data dictionary format, have at least one categorical variable. In addition, a special case of categorical data type is a Boolean data type and it was also not present in any of the element dictionaries. Both CDISC Define.XML and REDCap standards clearly distinguish categorical data elements and support enumeration of permissible values. For the purpose of data analysis and when data is loaded into a database, a categorical variable may still be implemented as a string or a number; however, a flag that indicates that only a set of permissible strings or numbers are expected as values represent good analytical practice.

By inspecting the data element title and description and study data, we found numerous categorical data elements; however, their formally listed data dictionary data type was not categorical and there was no flag marking them as categorical.

We consider string data type (sometimes also called character) to be a free-text entry with no restrictions. In other words, no set of permissible values is defined for a string data element. An example of a data element of type string (from study NCT01751646) is the element titled

**Table 5. Count and percentage of data elements by data type.**

| Data Type | Number of Data Elements | Percentage of Data Elements with a Type |
|---|---|---|
| String | 7569 | 48.6% |
| Numeric | 7444 | 47.8% |
| Date | 467 | 3.0% |
| Date-time | 93 | 0.6% |

'Reason missed vitamins—Specify' (data element ID: VTMRSP). It asks about the specific reasons why the patient missed taking vitamins. The study does not provide any list of permissible values for this question, and there are 107 distinct responses in the IPD data).

To demonstrate incorrect data type, we provide two examples. The first example shows the imperfect use of string data type. The study NCT00099359 contains a data element 'Severity grade of a medical event' (data element ID: GRADE). This categorical data element has five permissible values: 'mild', 'moderate', 'severe', 'life threatening' and 'death'. The IPD data shows 5 distinct values that are all subsumed by this permissible value set. An accurate data dictionary should adopt and use categorical data type as one of the valid data types. If, for some reason, this semantically-rich modelling approach for data dictionary is not used, it should at least use a flag or other mechanism to distinguish data as 'string-categorical' versus 'string-proper'.

The second example shows the other variant of this misclassification problem when a categorical data element is in the data dictionary marked as numeric data type (with numbers representing codes for a particular permissible value). As demonstrated in the previously cited retracted analysis [24], this misclassification can be even more error-prone. In our sample, an example of a numeric-categorical data element (a data element that is not formally designated as categorical in the data dictionary and not distinguished from numerical-proper by any flag) is from study NCT00005273 with title 'pain severity' that asks about severity of pain. It has the following permissible values: 0 for none, 1 for mild, 2 for moderate, 3 for severe, and 9 for unsure. Not interpreting this element as numerical value is crucial for correct data analysis.

If the data dictionary does not model correctly categorical variables, the provision of a complete permissible value dictionary can still fully compensate for the lack of distinction between numeric-categorical and numeric-proper or string-categorical and string-proper. However, we found that many studies in our sample did not provide the permissible value dictionary, and the problem can thus still occur.

**3.2.3 Data element description.** For data re-using researchers, the ability to properly interpret each data element is crucial. For that purpose, having an unambiguous description for each data element (in addition to a data element ID) facilitates this proper interpretation. If the data element definition or meaning cannot be fully understood (description is missing or is vague), the resulting analysis can misinterpret these data elements (leading to incorrect results) or exclude them from the analysis (leading to incomplete results).

We found missing data element descriptions in three studies. The element description field was missing for 4 447 DEs (16.9% out of all 26 328 DEs). One study (NCT00005274) with 4 249 DEs omitted descriptions for all its data elements, while two studies (NCT00865566 and NCT01233531) had missing descriptions for some DEs. The remaining 15 studies (83.3% out of 18 studies) provided descriptions for 100% of their DEs. We also do, however, acknowledge the fact that a well formulated data element name can be in some cases sufficient to fully define an element. In the aggregate DE file, we observed 327 DEs where the DE description was identical to the DE name (1.9% out of all 17 014 elements with descriptions). We also found that some DE descriptions were vague or difficult to understand. DEs with confusing description can sometimes be disambiguated by inspecting the IPD data. Although, such inspection may require the submission (and approval) of a formal data request that may not be necessary for the data dictionary alone.

**3.2.4 ClinicalTrials.gov record.** Review of the ClinicalTrials.gov records showed that 7 studies (38.9% out of 18 studies) provided study results. No study included a hyperlink (or actual file) for the data dictionary. Two studies (11.1%) referenced IPD availability on ClinicalTrials.gov.

### 3.3 Forms data dictionaries

Although it is theoretically possible for a study to not organize its DEs into groups, all 18 studies in our sample did group their elements and utilized a group ID mechanism. The majority of studies (11 out of 18 studies) used the case report form as the organizing principle (group ID is the form name). The remaining 7 studies organized their elements by study data table and used the table name as the group ID. The median number of forms in a study (or distinct groups) was 27 (ranging from a minimum of 2 (NCT02404311) to a maximum of 124 (NCT00005274).

A review of form names (or group IDs) can provide data re-using researchers a highly pragmatic and quick overview of what data was collected in a given study. If a standardized data collection instrument was used in a study, it may be easiest to discover it via the review of the forms dictionary. If a previously defined and standardized form addresses well the study's data collection needs, use of such a standardized form allows for a very straightforward method of meta-analysis across studies. For example, CDISC Clinical Data Acquisition Standards Harmonization standard (CDASH) defines such common forms for 'Protocol Deviations', 'Demographics', 'Adverse Event', 'End of Study', or 'Concomitant Medications' [26].

We identified a total of 28 common form names. Some form names, such as 'Off Study' (in 11 of 18 studies, 61.1.6%), 'Eligibility' (in 10 on 18 studies, 55.6%) and 'Visit Report' (in 10 of 18 studies, 55.6%) were very common in our set of studies. We defined very common as forms being present in 50% or more studies out of a set of studies that had a forms dictionary. Table 6 shows the very common form names and quantitative measures of their use (count of studies and percentage). Less common form names (present in two studies) were Body Measurements, Family History, Medical History, Missed Visit, Behavior, and Skin Test. Our results for common form names are affected by the fact that 10 (out of 18 studies) were executed by the same research network (Adolescent Medicine Trials Network for HIV/AIDS Interventions).

Similarly, to data elements, we have observed form names data inaccuracies in 11 studies within our sample. We observed form names (or group names) that were ambiguous, and instances of identical names used to refer to two clearly different forms (See Table 2 for an example: two forms both titled 'Specimen Tracking Form'). For data recipients, unambiguous and good data descriptions are important for proper analysis.

No study in our sample marked any of the forms as copyrighted. Copyright protection typically does not impact the use of the collected data in a secondary research analysis. However, for researchers obtaining common forms and data elements with the intent to use the most prominent ones in a future study (especially those used frequently in past studies), the copyright status is important.

### 3.4 Permissible values data dictionaries

Use of categorical data elements in research is extremely common and, as stated earlier, most studies would be expected to provide a permissible value dictionary.

Table 6. Very common form names and the number and percentage of studies their used in.

| Table definition | Count of Studies | Percentage of Studies |
|---|---|---|
| Off Study | 11 | 61.1% |
| Eligibility | 10 | 55.6% |
| Visit Report | 10 | 55.6% |
| Adverse Event | 9 | 50.0% |
| Diagnoses | 9 | 50.0% |

We were able to extract permissible values for 11 studies. The aggregate file contained a total of 7 669 permissible values for 1 815 DEs. The mean number of permissible values per data element is 4.23. Use of numerical IDs is most common: 1 511 categorical DEs (83.3%) used numbers as permissible values. The most frequent permissible value description was 'No' (for 1 334 DEs in 10 distinct studies).

In some cases, permissible values represent administered drugs (e.g., values 3TC, NFV, NVP and ZDV were permissible values for data element 'DRUGNM' which represents the drug name in trial NCT00099359). Permissible values can also represent laboratory tests. For example, 'CD4 T cell percent' is a permissible value for data element 'TESTNM' in study NCT01772823. Some permissible values can be further linked to established healthcare terminologies, such as RxNorm terminology for drugs or Logical Observation Identifiers Names and Codes (LOINC) terminology for laboratory tests. Both data dictionary standards (Define-XML and REDCap) allow for such annotation of permissible values by relevant external terminology codes. No study in our sample made any such annotations.

### 3.5 Studies using CDISC format

Six HIV studies we obtained for our analysis used a CDISC format. CDISC specifications mandate IPD data to be represented in an SDTM (Study Data Tabulation Model) format and a corresponding data dictionary in the Define-XML format. Table 7 lists the studies that used CDISC format (which are excluded from the main analysis). The SDTM format accommodates some data elements directly as a column in a standardized spreadsheet (we refer to those as SDTM-model-hardcoded DEs) and for other data elements it uses an entity-attribute-value (EAV) approach with concepts for those entities defined in CDISC Controlled Terminology (we refer to those as SDTM EAV DEs). CDISC Controlled Terminology (CT) thus represents yet a third, tightly linked and relevant CDISC standard. In fact, CDISC CT concepts are also used for coded permissible values. All six studies that used some CDISC standard used SDTM format and also provided a Define-XML dictionary.

## 4 Discussion

We obtained a large number of HIV trials and analyzed the format and content of their data dictionaries. To our knowledge, our study is the first to aggregate HIV data elements from a large set of completed HIV studies that were later shared via a platform or other mechanism.

The majority of studies used a custom, non-standardized format that required significant processing to make the data machine-readable. The lack of consensus to use a single standard should be viewed in light of the fact that data sharing of clinical study data is still a developing and evolving scientific challenge. Moreover, the studies in our sample may have been initiated

**Table 7. List of acquired studies using CDISC.**

| NCT_ID | Title |
|---|---|
| NCT03164564 | HPTN 084 Evaluating the Safety and Efficacy of Long-Acting Injectable Cabotegravir Compared to Daily Oral TDF/FTC for Pre-Exposure Prophylaxis in HIV-Uninfected Women |
| NCT01612169 | Project HOPE—Hospital Visit as Opportunity for Prevention and Engagement for HIV-Infected Drug Users |
| NCT00102349 | HIV and HCV Intervention In Drug Treatment Settings—1 |
| NCT00084175 | HIV/STD Safer Sex Skills Groups for Men in Drug Treatment Programs—1 |
| NCT00084188 | HIV/STD Safer Sex Skills Groups for Women in Drug Treatment Programs—1 |
| NCT01154296 | HIV Rapid Testing & Counseling in Sexually Transmitted Disease (STD) Clinics in the U.S. (Aware) |

at a time when the emphasis on proper data sharing (by research enterprise in general) was smaller. Only in recent years has the importance of study metadata become more prominent.

## 4.1 Related work

In the clinical domain of HIV, we did not find any prior study that analyzed HIV-specific research data elements. CDISC HIV therapeutic area user guide, published in January 2019, is the only relevant HIV-specific data element effort [3] (in addition to base SDTM elements, it highlights lab codes for CD4 count, LOINC codes for HIV viral load testing and mother-infant data linking among many other things). A study focused on data elements and data sharing for HIV registries was published by our team in 2019 [27]. There are, however, prior studies that are not specific to HIV and cover data elements and dictionaries for medicine in general. Sharma et al. analyzed three data dictionaries and used Archetype Modeling Language developed by the Clinical Information Modeling Initiative [28]. They also developed a platform (called D2Refine) for data element harmonization and standardization. In their report they discuss impediments to comparison and interoperability caused by the lack of standardization of data dictionaries from clinical studies. This lack of standardization we observed in our analysis as well. Another analysis by Strickler et al. analyzed case report forms and utilized CDISC Operational Data Model (ODM) [29] to represent data elements present on such forms. Finally, Tcheng et al. analyzed data elements from 32 registries in the 2018 project titled 'Common Healthcare Data Interoperability Project' (CHDIP) funded by The Pew Charitable Trusts in 2018) [30]. Their report defines 28 general CDEs (e.g., last name, date of death) and several composite CDEs (that group several elements; procedure history, medication history and laboratory result history).

## 4.2 Recommendations

Due to the evolution of some of the data dictionary standards and sharing policies, it is difficult to recommend a single data dictionary standard. As outlined in section 2.3, there are essentially two data dictionary standards (CDISC Define-XML and REDCap) to consider. Although, the REDCap format is not backed up by any formal Standard Developing Organization (SDO). A recommendation to adopt a CDISC Define-XML triggers the requirement to also adopt other related CDISC standards (including CDISC Controlled Term terminology and possibly CDISC SDTM) and such adoption requires significant training and technical expertise. We consider it too significant a hurdle to be universally recommending such an adoption. With regards to the REDCap format, we view it to be very similar to simply formulating a good set of best practices to create a single spreadsheet, compliant with FAIR principles [31, 32], that captures all significant parameters of all study data elements (either unique to the study or formal CDEs adopted by the study). In other words, if REDCap format is not formally utilized, we also find it acceptable to use an ad-hoc single spreadsheet-based data dictionary.

Based on transformation of data dictionaries and the creation of the aggregated DE file, we have designed a set of general recommendations for the optimal sharing of data element metadata for future studies in the HIV domain. We, however, do believe that these recommendations also apply beyond HIV and to other medical domains.

We enumerate some of the most important recommendations as a list below. We developed a more detailed set of recommendations as part of CONSIDER statement (Consolidated Recommendations for Sharing Individual Participant Data from Human Clinical Studies) to improve the practice of data sharing and reuse. It is available at w3id.org/CONSIDER and includes details defining the desired best data sharing practices, positive examples of studies

following the recommendations, challenging examples of studies where recommendations are not followed and notes on how to evaluate adherence of a study to a given recommendation (CONSIDER checklist).

- Provide data dictionary documentation separate from de-identified individual participant data. Since it contains no participant level data and it does not require local ethical approval as a condition of releasing the data dictionary (avoid a request wall for the data dictionary).

- Share a data dictionary as soon as possible. Do not wait until the data collection is complete.

- Provide the data dictionary in a single, machine-readable file.

- For each data element, provide a data type (such as numeric, date, string, categorical).

- For categorical data elements, provide a list of permissible values and distinguish when a numerical code or a string code is a code for a permissible value (versus when it is an actual number or string)

- Provide a complete data dictionary (all elements in the data are listed in a dictionary) and all types of applicable dictionaries (date elements, forms [or groupings], and permissible values). Utilize a description field, in addition to title, to fully describe a data element or form.

- At study design time and when resources allow, adopt previously defined applicable common data elements (including adoption of grouped data elements).

### 4.3 Grouped data elements

Within the analyzed data dictionaries, we saw instances of closely related DEs. For example, HIV 1 RNA viral load and HIV 1 RNA viral load date. This split into multiple self-standing DEs that are closely related data elements created higher counts of distinct data elements. In recent decades, several common data models (CDMs) for Electronic Health Record (EHR) data have emerged that group related data elements into more complex structures. Similarly, the CDISC SDTM standard provides higher level structures (LB [= laboratory] domain) that group data elements together into one row of data in a higher level structure. This grouping has been referred to as the EHR data convention [20] and is increasingly becoming the method of choice for modelling closely related DEs.

### 4.4 De-identification

Sharing of clinical study data is inherently linked to de-identification. If a study collected sensitive information (patient's exact date of birth or other identifying data enumerated in regulations or various policies), such information is removed or obfuscated prior to data sharing. Nine studies in our sample provided de-identification notes. Maintaining a data element dictionary can greatly facilitate IPD de-identification. For example, all elements with date data type may need a relative time obfuscation (shifting all events for a given participant by some fixed number of days). Annotation of data elements with common terminologies can help in finding data elements that need to be de-identified and it can help identify an appropriate de-identification method based on a knowledge base organized by CDE created from prior instances of study de-identification.

### 4.5 Limitations

Our findings are time dependent and based on studies shared at the time of our review (October 2019; 18 studies). Ongoing studies or studies currently being planned may use a more

complete data dictionary and employ better dictionary formats. Thanks to requirements to use CDEs included in new funding opportunity announcements [2], newly initiated studies are more likely to adopt CDEs during the study design stage. Because we searched specifically for HIV studies, our findings may not generalize fully to other clinical domains.

## 5 Conclusion

We analyzed 26 328 DEs from 18 HIV studies. All analyzed studies organized their DEs into groups. An average study, represented by the median of our set, had 427 data elements. The majority of studies used CSV files to share a data dictionary while 22% of the studies used a non-computable, PDF format. String and numeric data types were the most frequent data types with many studies incorrectly representing categorical data elements (100%) and not providing a full list of permissible values (39%). Only a minority of studies reported study results or linked IPD within their ClinicalTrials.gov record. We also identified two relevant data dictionary standards that have many features that encourage proper data sharing. Using our analysis and review, we designed a set of recommendations that provide best practices for data sharing for future studies.

## Acknowledgments

This research was supported by the NIH Office of AIDS Research and Intramural Research Program of the National Institutes of Health (NIH)/National Library of Medicine (NLM)/ Lister Hill National Center for Biomedical Communications (LHNCBC). We would like to thank Kin Wah Fung and Laritza Rodriguez for providing comments on the earlier version of the manuscript. The findings and conclusions in this article are those of the authors and do not necessarily represent the official position of NLM, NIH, or the Department of Health and Human Services.

## Author Contributions

**Conceptualization:** Nick Williams, Vojtech Huser.

**Data curation:** Craig S. Mayer.

**Formal analysis:** Craig S. Mayer, Vojtech Huser.

**Funding acquisition:** Vojtech Huser.

**Investigation:** Vojtech Huser.

**Methodology:** Craig S. Mayer, Nick Williams, Vojtech Huser.

**Project administration:** Craig S. Mayer, Vojtech Huser.

**Software:** Craig S. Mayer, Nick Williams, Vojtech Huser.

**Supervision:** Vojtech Huser.

**Validation:** Craig S. Mayer, Nick Williams, Vojtech Huser.

**Writing – original draft:** Craig S. Mayer, Vojtech Huser.

**Writing – review & editing:** Craig S. Mayer, Nick Williams, Vojtech Huser.

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
