## [Decision Letter · Decision Letter 0]

10 Jul 2020

PONE-D-20-13400

Analysis of Data Dictionary Formats of HIV Clinical Trials

PLOS ONE

Dear Dr. Mayer,

Thank you for submitting your manuscript to PLOS ONE. After careful consideration, we feel that it has merit but does not fully meet PLOS ONE’s publication criteria as it currently stands. Therefore, we invite you to submit a revised version of the manuscript that addresses the points raised during the review process.

We look forward to receiving your revised manuscript.

Kind regards,

Nguyen Tien Huy, Ph.D., M.D.

Academic Editor

PLOS ONE

Journal Requirements:

Reviewers' comments:

Reviewer's Responses to Questions

**Comments to the Author**

1. Is the manuscript technically sound, and do the data support the conclusions?

Reviewer #1: Yes

Reviewer #2: Partly

Reviewer #3: Yes

2. Has the statistical analysis been performed appropriately and rigorously? 

Reviewer #1: N/A

Reviewer #2: Yes

Reviewer #3: N/A

3. Have the authors made all data underlying the findings in their manuscript fully available?

Reviewer #1: No

Reviewer #2: Yes

Reviewer #3: No

4. Is the manuscript presented in an intelligible fashion and written in standard English?

Reviewer #1: Yes

Reviewer #2: Yes

Reviewer #3: Yes

5. Review Comments to the Author

Reviewer #1: The topic is important, as biomedical research in general needs to keep pressing on improving data sharing in a way that can enhance reproducible research.

A few comments:

1. It may help to expand somewhat on the reasons for standards of data sharing, e.g. reproducible research. The article jumps into the review without giving as much background as could be helpful.

2. If we think of the studies that were not included as part of the analyzed 18 studies as 'missing data', then this manuscript could do a more thorough job in explaining the sampling frame. Some expansion on the filtering profess from n studies to 18 studies, and additional information on reasons why some studies were not included (e.g., in a table) would help. If we want to make inference for a population of studies, then greater understanding of how the 18 studies are a biased sample of the population of studies would help.

3. The manuscript mentions dozens of data sharing platforms. I wonder about the level of diversity in the data standards across these platforms? Is part of the issue that the different platforms do not all adopt the same 'constitution' of data standards principles?

4. In some places, qualitative statements are made about studies having 100% of data elements with information included in a data dictionary, vs. < 100 of data elements with information included in a data dictionary. Is it possible to quantify this issue better, e.g. by providing the percentage instead of binary 100% vs. M< 100%?

5. Adding a few more tables and figures may help provide substantive results in a convenient manner to review and come back to.

6. I think the recommendations should be strengthened, by being more specific (e.g., to the data elements standards system) and complete, the recommendations will be the take home items of the manuscript I believe. Also, linking the work more to other standardization papers or documents would help.

Reviewer #2: it is a very interesting research and the creation of a single information model is a new idea, while there are some mistakes in using the words in improper scientific way and i recommend to enhance the language

Reviewer #3: The manuscript “Analysis of Data Dictionary Formats of HIV Clinical Trials” by Mayer et al. appeared to be an interesting analysis for data collection from the clinical trials of HIV studies. Their analysis and recommendations might be meaningful and helpful for data sharing in future original studies which could facilitate re-using data researchers for their works. However, I have some concerns that I think the authors should clarify them in their manuscript:

1. For the method, the authors said that they included and analyzed HIV studies, but HIV studies are not clear to me. Does it mean all studies with HIV population? For example, a study of another infectious disease which HIV is a comorbidity could be included?

2. Databases/Platforms/Networks that the authors searched to find included studies are not clear in the method. Although they presented several names of databases/platforms in the results, they are not all used ones. It makes the method hard for readers to repeat their steps. I recommend them to present all these databases/platforms/networks names in the method.

3. In the method, the mentioned sufficiently the definitions, but no analysis strategy was presented. That makes me feel hard to follow their results, especially the differentiation between the section Data element dictionaries and the section Data Element Description. In fact, I saw the overlapped results when they mentioned studies fully or partially missed data element description.

4. For the results, Page 8, line 186 – 187, the following sentence is confusing “One study from HPTN and the 5 studies acquired from the NIDA Data Archive used a CDISC format, which we analyzed separately.” At first, I think that they will analyzed these studies separately from 18 other studies. But no analysis for these studies was done, this meant they are excluded. The authors should rewrite the sentence.

6. PLOS authors have the option to publish the peer review history of their article (what does this mean?). If published, this will include your full peer review and any attached files.

Reviewer #1: No

Reviewer #2: No

Reviewer #3: **Yes: **Dao Ngoc Hien Tam

---

## [Author Response · Author response to Decision Letter 0]

3 Aug 2020

Also available as an attached file:

Dear Dr. Nguyen Tien Huy,

We thank you and the reviewers for the input and constructive comments on our manuscript. We have thoroughly reviewed each comment and our manuscript and have edited our paper taking into account each comment. We are now submitting the edited and improved version of our manuscript.

See below our response (in bold font) to each reviewer comment. Our response follows the reviewer’s comments (shown in regular un-bolded text).

To ensure transparency and reproducibility all underlying data used in our analysis can be found in our aggregated data element file at our github repository at https://github.com/lhncbc/CDE/tree/master/hiv/datadictionary

Thank you,

Craig Mayer, Nick Williams and Vojtech Huser

Reviewer #1:

The topic is important, as biomedical research in general needs to keep pressing on improving data sharing in a way that can enhance reproducible research.

We greatly appreciate each of the comments and we considered each one and made many additions, specified below, to improve the ability of articulating the papers meaning and results.

A few comments:

1. It may help to expand somewhat on the reasons for standards of data sharing, e.g. reproducible research. The article jumps into the review without giving as much background as could be helpful.

We added a paragraph in the middle of the introduction section. The added text expands on the usefulness of CDEs, data standards, good data sharing practices. The added text also comments on the use of such standards in improving the re-usability of the data and the use of existing tools and techniques.

2. If we think of the studies that were not included as part of the analyzed 18 studies as 'missing data', then this manuscript could do a more thorough job in explaining the sampling frame. Some expansion on the filtering profess from n studies to 18 studies, and additional information on reasons why some studies were not included (e.g., in a table) would help. If we want to make inference for a population of studies, then greater understanding of how the 18 studies are a biased sample of the population of studies would help.

We have revised section 3.1 based on this comment. We expanded on our filtering and exclusion criteria in results section 3.1. We stated that we excluded studies that did not provide a data dictionary in a machine-readable format or documents that can be converted into a machine readable format due to our analysis strategy.

3. The manuscript mentions dozens of data sharing platforms. I wonder about the level of diversity in the data standards across these platforms? Is part of the issue that the different platforms do not all adopt the same 'constitution' of data standards principles?

Our observation was that only NIDA Data Share had any globally used data standard, while the other platforms allowed studies to use whatever format was preferred. The revised manuscript has an added a sentence stating this in section 3.1 of the results.

4. In some places, qualitative statements are made about studies having 100% of data elements with information included in a data dictionary, vs. < 100 of data elements with information included in a data dictionary. Is it possible to quantify this issue better, e.g. by providing the percentage instead of binary 100% vs. M< 100%?

Thank you for the comment and a good new idea. We agree that quantifying this statement regarding the number of data elements in the the data vs. data elements in the data dictionary is desirable. We have conducted an additional analysis and modified the manuscript. We used a 50% sample of studies that could be used for this investigation (where we have both data dictionary (DD) and individual participant data [IPD]). The revised manuscript now has a new new Table 4 with results of this analysis and added discussion of results. (We no longer just report about 100% vs. <100% but provide exact numbers for all studies we analyzed for this indicator. See revised 3.2.1. The new table clearly shows that the completeness of DD ranges between 45.4% to 100%. The low value of 45.4% is an outlier. Four studies out of five analyzed, had completeness >85%.

5. Adding a few more tables and figures may help provide substantive results in a convenient manner to review and come back to.

To improve the readability of the results, we added four new tables. The first is in response to the previous comment showing the percentage of data elements in the data dictionary (table 4). The second is the breakdown of data elements by data type (Table 5). The third is the list of very common form names (Table 6), and the fourth is the list of trials using the CDISC standard (Table 6).

6. I think the recommendations should be strengthened, by being more specific (e.g., to the data elements standards system) and complete, the recommendations will be the take home items of the manuscript I believe. Also, linking the work more to other standardization papers or documents would help.

We are glad that reviewer#1 has the same stance on standardization (that it should be strengthened). In the revised manuscript, we have expanded section 4.2 (Recommendations). In the revised introduction, we also refer to 5 new references in a revised Introduction section to refer to relevant papers arguing for CDEs and advanced data sharing. The revised recommendation section 4.2 now comments specifically on the standard choice and also links to 2 additional new references.

Reviewer #2:

It is a very interesting research and the creation of a single information model is a new idea, while there are some mistakes in using the words in improper scientific way and I recommend to enhance the language.

We carefully reread the manuscript and made revisions where we encountered improper scientific language. (e.g. changes to Introduction and Trial Acquisition sub-sections of both Methods and Results sections). We made additional edits to improve the language and better explain the reasons for certain term usage. Additional revisions to correct “improper scientific language” were made in response to reviewer #1 and reviewer #3. 

Reviewer #3:

 The manuscript “Analysis of Data Dictionary Formats of HIV Clinical Trials” by Mayer et al. appeared to be an interesting analysis for data collection from the clinical trials of HIV studies. Their analysis and recommendations might be meaningful and helpful for data sharing in future original studies which could facilitate re-using data researchers for their works. However, I have some concerns that I think the authors should clarify them in their manuscript:

Thank you for your comments and input, as we have considered them and edited our paper as stated below to incorporate the comments and answer any concerns.

1. For the method, the authors said that they included and analyzed HIV studies, but HIV studies are not clear to me. Does it mean all studies with HIV population? For example, a study of another infectious disease which HIV is a comorbidity could be included?

We clarified this statement by adding to section 2.1 in the methods where we explained that HIV studies describes any study with HIV positive patients or any study relating to the contracting HIV, such as HIV vaccines and prevention. 

2. Databases/Platforms/Networks that the authors searched to find included studies are not clear in the method. Although they presented several names of databases/platforms in the results, they are not all used ones. It makes the method hard for readers to repeat their steps. I recommend them to present all these databases/platforms/networks names in the method.

We have modified the manuscript and added the list of all searched platforms (see the methods in sub-section 2.1). This should help anyone who is trying to reproduce our search results. The revised expanded text also describes which sources we acquired data from and which sources did not have any studies that we included.

3. In the method, the mentioned sufficiently the definitions, but no analysis strategy was presented. That makes me feel hard to follow their results, especially the differentiation between the section Data element dictionaries and the section Data Element Description. In fact, I saw the overlapped results when they mentioned studies fully or partially missed data element description.

Thank you for the input. We agreed that this information regarding missing data description was repeated and so we removed it from the section data element dictionary (section 3.2) and expanded on it in data element description (section 3.2.3). We also added more specific sentences regarding our analysis techniques in the methods, which can be found throughout the revised section 2.2.

4. For the results, Page 8, line 186 – 187, the following sentence is confusing “One study from HPTN and the 5 studies acquired from the NIDA Data Archive used a CDISC format, which we analyzed separately.” At first, I think that they will analyzed these studies separately from 18 other studies. But no analysis for these studies was done, this meant they are excluded. The authors should rewrite the sentence.

We clarified this sentence to articulate that these CDISC studies were separated and not included in any analysis with the other studies, with their presence being the result itself. The emphasis of the paper was to learn mostly from the ad-hoc data dictionaries rather than simply re-discover CDISC standard benefits. In our future work (that is not limited to HIV clinical domain), we hope to collect as large as possible sample of CDISC-formatted studies and analyze those in a future publication.

---

## [Decision Letter · Decision Letter 1]

18 Sep 2020

Analysis of data dictionary formats of HIV clinical trials

PONE-D-20-13400R1

Dear Dr. Mayer,

We’re pleased to inform you that your manuscript has been judged scientifically suitable for publication and will be formally accepted for publication once it meets all outstanding technical requirements.

Kind regards,

Nguyen Tien Huy, Ph.D., M.D.

Academic Editor

PLOS ONE

Additional Editor Comments (optional):

Reviewers' comments:

Reviewer's Responses to Questions

**Comments to the Author**

1. If the authors have adequately addressed your comments raised in a previous round of review and you feel that this manuscript is now acceptable for publication, you may indicate that here to bypass the “Comments to the Author” section, enter your conflict of interest statement in the “Confidential to Editor” section, and submit your "Accept" recommendation.

Reviewer #1: All comments have been addressed

Reviewer #3: All comments have been addressed

2. Is the manuscript technically sound, and do the data support the conclusions?

Reviewer #1: Yes

Reviewer #3: Yes

3. Has the statistical analysis been performed appropriately and rigorously? 

Reviewer #1: Yes

Reviewer #3: Yes

4. Have the authors made all data underlying the findings in their manuscript fully available?

Reviewer #1: Yes

Reviewer #3: Yes

5. Is the manuscript presented in an intelligible fashion and written in standard English?

Reviewer #1: Yes

Reviewer #3: Yes

6. Review Comments to the Author

Reviewer #1: Thank you for the revision which was responsive. I especially appreciate the addition of tables that provide greater quantification of the data issues.

Reviewer #3: (No Response)

7. PLOS authors have the option to publish the peer review history of their article (what does this mean?). If published, this will include your full peer review and any attached files.

Reviewer #1: No

Reviewer #3: No

---

## [Editor Report · Acceptance letter]

24 Sep 2020

PONE-D-20-13400R1 

Analysis of data dictionary formats of HIV clinical trials 

Dear Dr. Mayer:

I'm pleased to inform you that your manuscript has been deemed suitable for publication in PLOS ONE. Congratulations! Your manuscript is now with our production department. 

Kind regards, 

on behalf of

Dr. Nguyen Tien Huy 

Academic Editor

PLOS ONE